Dinophiliformia early neurogenesis suggests the evolution of conservative neural structures across the Annelida phylogenetic tree

Fofanova Elizaveta 1 fofanova@idbras.ru
Mayorova Tatiana D. 1 2
Voronezhskaya Elena E. 1
1 Department of Comparative and Developmental Physiology, Koltzov Institute of Developmental Biology RAS , Moscow , Russia
2 Laboratory of Neurobiology, National Institute of Neurological Disorders and Stroke , Bethesda, Maryland , USA
Ferrier David
Electronic publication date: 2021 Dec 8
Publication date: 2021
Volume: 9
Electronic Location ID: e12386
Received 2021 May 13; Accepted 2021 Oct 4
Copyright: © 2021 Fofanova et al.
Copyright year: 2021
Copyright holder: Fofanova et al.
License: This is an open access article distributed under the terms of the Creative Commons Attribution License, which permits unrestricted use, distribution, reproduction and adaptation in any medium and for any purpose provided that it is properly attributed. For attribution, the original author(s), title, publication source (PeerJ) and either DOI or URL of the article must be cited.
License URL: https://creativecommons.org/licenses/by/4.0/

Keywords: Neurogenesis, Pioneer neurons, Trochophore, Lophotrochozoa, Serotonin, Apical organ, Ventral nerve cord, FMRFamide, Dinophiliformia, Annelida

Funding: Russian Foundation for Basic Research grant 19-34-60040 Russian Science Foundation grant 17-14-01353 The reported study was funded by the Russian Foundation for Basic Research, grant # 19-34-60040. Experiments with 5-HT and 5-HTP incubations at the early developmental stage were done in the frame of Russian Science Foundation grant # 17-14-01353. The funders had no role in study design, data collection and analysis, decision to publish, or preparation of the manuscript.

==============================
Despite the increasing data concerning the structure of the adult nervous system in various Lophotrochozoa groups, the early events during the neurogenesis of rare and unique groups need clarification. Annelida are a diverse clade of Lophotrochozoa, and their representatives demonstrate a variety of body plans, lifestyles, and life cycles. Comparative data about the early development are available for Errantia, Sedentaria, Sipuncula, and Palaeoannelida; however, our knowledge of Dinophiliformia is currently scarce. Representatives of Dinophiliformia are small interstitial worms combining unique morphological features of different Lophotrochozoan taxa and expressing paedomorphic traits. We describe in detail the early neurogenesis of two related species: Dimorphilus gyrociliatus and Dinophilus vorticoides, from the appearance of first nerve cells until the formation of an adult body plan. In both species, the first cells were detected at the anterior and posterior regions at the early trochophore stage and demonstrated positive reactions with pan-neuronal marker anti-acetylated tubulin only. Long fibers of early cells grow towards each other and form longitudinal bundles along which differentiating neurons later appear and send their processes. We propose that these early cells serve as pioneer neurons, forming a layout of the adult nervous system. The early anterior cell of D. vorticoides is transient and present during the short embryonic period, while early anterior and posterior cells in D. gyrociliatus are maintained throughout the whole lifespan of the species. During development, the growing processes of early cells form compact brain neuropile, paired ventral and lateral longitudinal bundles; unpaired medial longitudinal bundle; and commissures in the ventral hyposphere. Specific 5-HT- and FMRFa-immunopositive neurons differentiate adjacent to the ventral bundles and brain neuropile in the middle trochophore and late trochophore stages, i.e. after the main structures of the nervous system have already been established. Processes of 5-HT- and FMRFa-positive cells constitute a small proportion of the tubulin-immunopositive brain neuropile, ventral cords, and commissures in all developmental stages. No 5-HT- and FMRFa-positive cells similar to apical sensory cells of other Lophotrochozoa were detected. We conclude that: (i) like in Errantia and Sedentaria, Dinophiliformia neurogenesis starts from the peripheral cells, whose processes prefigure the forming adult nervous system, (ii) Dinophiliformia early cells are negative to 5-HT and FMRFa antibodies like Sedentaria pioneer cells.

Introduction

Annelida are a diverse group of organisms with a variety of adult and larval body plans. Recent studies on neurogenesis have been focused on the species having free-swimming larvae in their life cycle: Sedentaria (C. teleta, M. fuliginosus); Errantia (P. dumerilii, P. maculata), Palaeoannelida (O. fusiformis), and Sipuncula (T. lageniformis). Detailed data concerning the neurogenesis of Chaetopteriformia, Amphinomida, and Dinophiliformia are absent.

Our study is focused on early neurogenesis events in two species of the group Dinophiliformia, small interstitial worms combining unique morphological features of different Lophotrochozoan taxa and expressing paedomorphic traits. Dinophiliformia have segmented epithelial structures like other Annelida, although they have no chaeta; they possess parenchymatous organization and protonephridia similar to Platyhelminthes; and utilize gliding ciliary locomotion like meiobentic invertebrates and mollusks. Representatives of Dinophiliformia have direct development inside the egg capsule packed in cocoon and hatch as small worms; however, they possess specific ciliary structures (prototroch, ventral ciliary field) similar to free-swimming larvae (Beklemishev & Kabata, 1969; Fofanova, Nezlin & Voronezhskaya, 2014; Kerbl et al., 2016a). Moreover, the adult species resemble a trochophore larva in that they maintain prototroch, ciliary bands, ventral ciliary field, and protonephridia. Such characteristics allows to call the adult species of Dinophiliformia “neotenic trochophore” (Beklemishev & Kabata, 1969). A later study of Dinophiliformia demonstrated paedomorphic traits in the adult nervous, muscular, and ciliary systems and supported the progenetic origin of Dinophiliformia (Windoffer & Westheide, 1988; Kerbl et al., 2016b). From one point of view, an interstitial environment (bottom marine environment or meiobenthic) may drive the paedomorphic evolution (Westheide, 1987), while another author suggests that interstitial organisms first appeared in evolution (Boaden, 1989). In both cases, the paedomorphic animals demonstrate simplified features that probably correspond to the early ontogenetic stages of an ancestral taxon. Moreover, the nervous system is one of the most conservative structures, the comparative analysis of which may infer the basal plan of the last common ancestor. That is why the structure of the Dinophiliformia nervous system has attracted the particular attention of researchers (Fofanova & Voronezhskaya, 2012; Fofanova, Nezlin & Voronezhskaya, 2014; Kerbl et al., 2016a, 2016b, 2017). However, the specification of the earliest differentiating cells expressing neuronal phenotype, the arrangement of their processes, and their fate in the course of development have not been addressed in these animals.

In a variety of Lophotrochozan species, the first neurons appear as early as the trochophore stage during development (Schmidt-Rhaesa, Harzsch & Purschke, 2015). They are often located peripherally, and their emanating processes establish the network resembling the anlagen of the developing adult (definitive) nervous system. Therefore, in some species of mollusks and annelids, the early peripheral cells were speculated to serve the function of pioneer neurons (Croll & Voronezhskaya, 1996; Voronezhskaya, Tyurin & Nezlin, 2002; Voronezhskaya & Elekes, 2003; Voronezhskaya & Ivashkin, 2010; Nezlin & Voronezhskaya, 2017; Yurchenko et al., 2019; Kumar et al., 2020). Pioneer neurons were first described in insect development and represent the cells whose processes navigate or pioneer the growing axons of later differentiating neurons (Bate, 1976; Klose & Bentley, 1989).

Among Lophotrochozoa, the first neurons demonstrate a positive reaction to serotonin antibodies (5-HT-IR) in Annelida and Nemertea (Voronezhskaya, Tsitrin & Nezlin, 2003; Fischer, Henrich & Arendt, 2010; Chernyshev & Magarlamov, 2010), while in mollusks they are positive to anti-FMRFamide antibodies (FMRFa-IR) (Croll & Voronezhskaya, 1996; Dickinson, Croll & Voronezhskaya, 2000; Voronezhskaya, Tyurin & Nezlin, 2002; Voronezhskaya, Tsitrin & Nezlin, 2003; Dickinson & Croll, 2003). These researches used a limited number of markers: acetylated alpha-tubulin, serotonin (5-HT), or FMRFamide (FMRFa) antibody to visualize the location and morphology of the earliest nerve cells. Until now, no specific markers were found for the early peripheral cells apart from the generally used immunolabelling (Conzelmann & Jékely, 2012; Kumar et al., 2020).

In addition to the early peripheral neurons, the other nerve elements were found in the trochophore stage, which belong to the apical or aboral sensory organ (ASO) of the larvae (Lacalli, 1981, 1994; Page & Parries, 2000; Page, 2002; Nielsen, 2005, 2004). The ASO is a part of the larval nervous system, located at the anterior pole of larvae, and comprises an apical ciliary tuft and receptor cells. In most representatives of Lophotrochozoa, the ASO consists of a specific number of flask-shaped receptor cells and displays serotonin-like immunoreactivity, and sometimes also FMRFamide-like immunoreactivity. The long basal processes of apical cells form a compact apical neuropile (Richter et al., 2010). While Dinophiliformia belongs to the lophotrochozoan Annelida, no ASO sign has been mentioned in their representatives.

Typically, in the course of development, the other elements of the larval nervous system (prototroch nerve, hyposphere nerve ring, etc.) and the anlagen of the adult nervous system (cerebral ganglia, ventral nerve cords, esophageal nerve circle, etc.) emerge soon after the appearance of the early peripheral and ASO neurons in most Lophotrochozoa (Nezlin, 2010; Hejnol & Lowe, 2015; Nezlin & Voronezhskaya, 2017; Yurchenko et al., 2019; Kumar et al., 2020).

Dinophiliformia includes three clades: Lobatocerebrum, Dinophilus, and Dimorphilus (Martín-Durán et al., 2021; Worsaae et al., 2021). We chose Dinophilus vorticoides and Dimorphilus gyrociliatus to analyze the cells, which express positive immunoreaction against a pan-neural marker–acetylated α-tubulin, in combination with immunoreaction to specific neuronal markers–5-HT and FMRFamide, during these worms’ development inside the egg capsule. We emphasize the time of appearance and location of the early peripheral cells, their fate, and the path of their processes; we analyze the relation between the early peripheral cells and the cells differentiating within the structures of the forming adult nervous system. We also searched for cells expressing the ASO phenotype in both normal and experimental conditions of increased 5-HT synthesis. Our work presents a detailed description of the neural cells from the first appearance until the formation of the main structures of the adult nervous system.

Materials and Methods

Culture maintaining

The Dimorphilus gyrociliatus culture was obtained from the Mediterranean Sea, Napoli Zoological Station (Italy). The animals were reared in small plastic aquaria with artificial seawater (33‰ salinity) at 21 °C without aeration and fed with homogenized frozen nettle (Urtica sp.) leaves once a week. The worms lay cocoons on the substrate, the wall, or the base of the plate. During daily water change, we collected all the cocoons and put them into fresh vials. Thus, we obtained the dated developmental stages from cleavage to pre-hatch specimens. D. gyrociliatus cocoons contained 1–9 large (female) representatives and several small (dwarf male) representatives (Shearer, 1911; Mauri, Baraldi & Simonini, 2003). In our work, we studied only females of D. gyrociliatus.

The work on Dinophilus vorticoides was conducted during the summer seasons at the White Sea, Pertsov White Sea Biological Station. The worms were collected in a subtidal zone during a low tide. We kept them in small tanks without aeration and six-well plates (for working with adhesive cell cultures) at 10 °C in filtered seawater and fed them with diatom Phaeodactylum sp. and Pseudinicshea delicatissima every day. Water was changed every time before feeding. The cocoons of D. vorticoides contained up to 30 representatives of the same size and both sexes.

Both Dinophiliformia species laid cocoons regularly. The cocoons were gently removed from the aquaria using a glass Pasteur pipette and placed into 30 mm Petri dishes with filtered seawater. The egg capsules were mechanically extracted from the cocoon and fertilization envelope with tiny needles for further immunocytochemical procedures. Dinophilus and Dimorphilus curl up after the gastrula stage with the ventral side out during the further early-late trochophore stages. Therefore, our images often show curved specimens, especially during the late stages.

Immunocytochemistry

D. gyrociliatus and D. vorticoides at various developmental stages were subjected to whole-mount immunostaining using standard protocols and a range of markers. Each staining was carried out using at least 40–50 specimens of each developmental stage. Antibodies against the neurotransmitters serotonin (5-HT), FMRFamide (FMRFa), and acetylated alpha-tubulin, as a neuronal, pan-neuronal, and ciliary marker, were used together with confocal laser scanning microscopy. This set of antibodies has been successfully applied to investigate neurogenesis in a variety of invertebrate larvae. Antibodies against acetylated tubulin, the neuropeptide RFamide, and the monoamine transmitter serotonin label individual neurons and their processes, as well as tubulin-containing ciliary structures (Schmidt-Rhaesa, Harzsch & Purschke, 2015). The specificity of the antibodies used in our study has previously been shown in various representatives of lophotrochozoan taxa including mollusks and annelids (Voronezhskaya & Elekes, 1996; Dickinson, Croll & Voronezhskaya, 2000; Voronezhskaya, Tsitrin & Nezlin, 2003; Wanninger et al., 2005; Kristof, Wollesen & Wanninger, 2008; Dyachuk & Odintsova, 2009; Fischer, Henrich & Arendt, 2010; Carrillo-Baltodano et al., 2019; Kumar et al., 2020; Carrillo-Baltodano et al., 2021).

The representatives of Dinophilus vorticoides and Dimorphilus gyrociliatus at different developmental stages were fixed with 4% paraformaldehyde (PFA) in phosphate-buffered saline (PBS, 0.01 mM, pH = 7.4) at 4 °C overnight. After fixation, the samples were washed three times at 15-min intervals in PBS and incubated in a blocking solution (10% normal goat serum and 1% bovine serum albumin in PBS) for 30 min at room temperature to eliminate non-specific binding. Then, preparations were incubated with primary antibodies for 3 days at 10 °C. We used anti-acetylated α-tubulin antibodies (Sigma–Aldrich, Burlington, MA, USA; T-6793; mouse; monoclonal: clone 6-11B-1, ascites fluid) diluted 1:5,000–1:10,000 in PBS containing 0.1% Triton X-100 (PBS-TX) to label stabilized microtubules in ciliary bands and neural elements. Specific neural cells were marked with anti-5HT and anti-FMRFamide antibodies (Immunostar, Hudson, WI, USA; 428002; rabbit; polyclonal; Product ID: 20080 and Immunostar, Hudson, WI, USA; 410002; rabbit; polyclonal; Product ID: 20091). Primary antibodies were washed three times for 10 min in PBS-TX solution. The primary antibodies were visualized with respective secondary goat-anti-rabbit and goat-anti-mouse antibodies conjugated with Alexa-488 (1:1,000, Molecular Probes, Eugene, OR, USA; A-11008; goat; polyclonal) and Alexa-633 (1:1000, Molecular Probes, Eugene, OR, USA; A-21050; goat; polyclonal) diluted in PBS containing 0.1% Triton X-100 overnight at 10 °C. Then, the preparations were washed with PBS three times for 10 min. Replacement of the primary antibodies with serum resulted in no staining. Reversal of the secondary antibodies (Alexa 633 and Alexa 488) gave the same staining pattern.

Cell nuclei were stained with DAPI (0.25 µg/ml) during the second washing. After washing in PBS, the specimens were mounted on glass slides in 70% glycerol for microscopic analysis and image acquisition. The elements demonstrating positive reactions were considered 5-HT and FMRFa immunoreactive (5-HT-IR and FMRFa-IR, respectively).

Confocal scanning microscopes TCS-SPE, TCS-SP5 (Leica Microsystems, Wetzlar, Germany), and Nikon A1 (Nikon, Tokyo, Japan) equipped with the appropriate set of lasers, filters, and detectors were used for the detailed study of the various developmental stages. Stacks of optical sections taken with ×63 objectives and 0.3–0.5 μm intervals were processed with Leica LAS AF (Leica Microsystems, Wetzlar, Germany) and Image J (NIH, Stapleton, NY, USA) to obtain two-dimensional images. For each sample, the optimal number of stacks was selected to demonstrate the structures of interest. Optical sections were projected onto one image and then imported into the Adobe Photoshop CC; the only parameters to be changed were brightness and contrast.

Application of 5-HT and its biochemical precursor 5-HTP

D. vorticoides at different developmental stages were incubated in 10−5 M serotonin (5-HT) and 5-hydroxytryptophan (5-HTP) (Sigma–Aldrich, Burlington, MA, USA) for 3 h. Fresh stock solutions in distilled water (10−2 M) were prepared immediately before each experiment. Negative controls for each experiment included treatment of animals with the final concentration of vehicle (distilled water) in filtered seawater. The experiments were performed at 10 °C. After the incubation, the animals were washed with filtered seawater four times and were then fixed and stained with anti-5-HT and anti-tubulin antibodies as described above.

Results

Relative staging

Duration of development within the egg capsule differs in Dimorphilus gyrociliatus and Dinophilus vorticoides. D. gyrociliatus hatch after 5 days, whereas D. vorticoides hatch approximately 2 weeks after oviposition. External ciliary structures develop sequentially in the anterior-to-posterior direction in both species. Each stage has a specific ciliary pattern. Hence, we used external ciliation as a marker for the relative staging of D. vorticoides and D. gyrociliatus development.

The first cilia are visualized soon after stomodeum emergence when the gastrula stage can be distinguished in both species. These cilia appear in the episphere and constitute a prototroch (Fig. 1A). The prototroch represents a circular ciliary band growing from above the stomodeum toward the dorsal side; since the left and right branches of the prototroch do not reach each other on the dorsal side, the prototroch remains open there.

Figure 1 External ciliation, relative staging and developmental timing in two dinophilid species.

Upper row–ventral view on wholemounts with external ciliation revealed by anti-acetylated alpha-tubulin labeling. Lower row–schematic representation of the key ciliary structures. There are three developmental stages in Dimorphilus gyrociliatus (D. g.) and Dinophilus vorticoides (D. v.): early trochophore (A), middle trochophore (B), and late trochophore stages (C). (A) The prototroch development marks the early trochophore stage. (B) The ventral ciliary field developing in the anterior-to-posterior direction characterizes the middle trochophore stage. (C) The acrotroch developing in the episphere and ciliary bands forming in the anterior-to-posterior direction characterize late trochophore stage. hpo–hours post oviposition, dpo–days post oviposition. Note substantial difference in the rate of development of the two species. Abbreviations: act–acrotroch, cb–ciliary bands, pt–prototroch; vcf–ventral ciliary field. Scale bar–30 μm.

When the prototroch has been developed, the ventral ciliary field appears along the midline of the ventral hyposphere beneath the mouth opening. The ventral ciliary field elongates in the anterior-to-posterior direction (Fig. 1B) and becomes broader and bushier during further development; its cilia become longer.

When the ventral ciliary field has been established, transversal ciliary bands start to grow. One ciliary band, called the acrotroch, appears in the episphere above the prototroch. Like the prototroch, the acrotroch extends to the dorsal side and forms an incomplete ring around the episphere open on the dorsal side. Other ciliary bands emerge in the hyposphere, i.e. under the prototroch. Initially, they look like paired narrow ciliated strip-shaped regions on both sides of the ventral ciliary field. During further development, the ciliary bands become broader and longer as they extend to the dorsal side, where they fuse, forming ring-shaped ciliary bands around the hyposphere (Fig. 1C). The number of the ciliary bands increases in the anterior-to-posterior direction, reaching five by the moment of hatching (Fig. 1C).

D. gyrociliatus and D. vorticoides are direct developers and hatch as the small young juvenile worm. Nevertheless, the stages inside the egg capsule are similar to those that pass the free-swimming larvae of other Lophotrochozoa. The characteristic ciliated structures such as prototroch and ventral ciliary field consequently appear in the course of both dinophlid species’ development. Prototroch is the main characteristic of a typical trochophore according to Rouse and Nielsen’s definition (Rouse, 1999, 2000). Based on the spatio-temporal ciliation pattern, we suggest the following postgastrulation developmental stages in D. vorticoides and D. gyrociliatus: early trochophore, middle trochophore and late trochophore (Figs. 1A–1C).

Antibodies against acetylated α-tubulin mark most neuronal cell bodies and their growing fibers. Cross-staining with antibodies against acetylated α-tubulin and the neurotransmitters serotonin and FMRFamide provides a visualization of specific neurons and their processes. We provide a description of the neuronal pattern in two Dinophiliformia species at each defined developmental stage.

Neurogenesis during the early trochophore stage

In D. gyrociliatus and D. vorticoides, the first specific elements expressing a positive reaction to anti-tubulin antibodies occurred during the early trochophore stage. In both species, the first cells demonstrate neither 5-HT- nor FMRFa- immunoreactivity (Fig. 2), so the structures described at this stage are only detected with anti-tubulin antibodies.

Figure 2 The first nerve elements appear at the early trochophore stage in Dimorphilus gyrociliatus and Dinophilus vorticoides.

Yellow–acetylated tubulin immunoreactivity; magenta–5-HT-like immunoreactivity; cyan–FMRFa-like immunoreactivity; red–DAPI. Ventral view. The apical pole is up. A–F′–Dimorphilus gyrociliatus; G–M–Dinophilus vorticoides. (A) The early anterior cell (asterisk) differentiate in the episphere with initially two symmetrical lateral processes growing towards the prototroch; (B) Later on, this cell extends multiple processes. In the hyposphere, these processes construct the anlagen of ventral (vb) and lateral bundles (lb) and the first commissure (c). In the episphere, they form a compact brain neuropile (np). Two symmetrically located cells emerge on the left and right sides of the stomodeum (frames); (C) Neither 5-HT nor FMRFa immunoreactivity was detected; (D, E) Nuclear staining confirms that tubulin immunolabelling is located to the body of the solitary anterior cell (F, F') High magnification of solitary left and right tubulin-positive ventral cells framed in B. (G) The early anterior and posterior cells (asterisks) with short emanating fibers develop at the opposite poles of the embryo; (H) Soon after that, the processes of the early cells form the anlagen of the neuropile (np), paired ventral bundles (vb), and two commissures (c). The body of the anterior cell is no longer visible. Symmetrical cell bodies develop at the base of each commissure (double arrows); (I) Neither 5-HT nor FMRFa immunoreactivity was detected; (J) The early solitary anterior cell framed in G; (K) The body of the cell at the base of a commissure and its processes joining the ipsilateral ventral nerve bundle; (L, M) Tubulin-immunoreactivity in the body and processes of the early posterior cell. Abbreviations: arrowheads indicate protonephridia, c–commissure, ln–lateral bundles, n–nucleus, np–neuropile, pt–prototroch, vb–ventral bundles. Scale bars: A–C, F–H–25 μm; D, E, L, M–1 μm; F, F′, J–2.5 μm; K–3.5 μm.

In D. gyrociliatus, the early trochophore stage passes relatively fast and lasts for just 1 day (during the second day after oviposition). So, during this period, the visualized structures vary significantly in different samples. Therefore, the sequence of neurogenesis was defined only after the analysis of a large number of samples.

In D. gyrociliatus, the first cell appears at the periphery on the ventral side in the episphere, right above the mouth (Fig. 2A). This early anterior cell has two symmetrical basal processes which grow laterally towards the prototroch (Fig. 2A). Nuclear staining confirms that the positive immunoreaction occurs in the body of the single solitary anterior cell but not in the neighboring cells (Fig. 2D).

By the end of the early trochophore stage, a pair of protonephridia develop in the episphere and are clearly visible as tubulin-positive structures in addition to tubulin-positive fibers of early cells (Fig. 2B, double arrowheads). The processes of the early anterior cell extend further in the posterior direction, following the curvature of the developing specimen, and form the anlagen of the lateral and ventral bundles (Fig. 2B). Growth cones are visible at the end of the left and right ventral growing fibers. Some of the posterior processes grow towards each other, cross midline, and form the first commissure (Fig. 2B). In addition to the anterior cell, two symmetrically located cells emerge on the left and right sides of the stomodeum (Fig. 2B). Anteriorly oriented branches of anterior and lateral cells bend towards each other and form the brain neuropile (Fig. 2B). Nuclear staining confirms that no additional cell bodies with positive immunoreaction occur in the anterior region or laterally to the stomodeum (Figs. 2E, 2F, 2F′).

No 5-HT-IR and FMRFa-IR elements were detected by the end of the early trochophore stage (Fig. 2C).

In D. vorticoides, the early trochophore stage lasts 3 days (5–7 days after oviposition). Two tubulin-positive cells are distinguished on the anterior and posterior poles simultaneously on the fifth day. The anterior cell is located ventrally, while the posterior cell occupies the dorsal side of the hyposphere (Fig. 2G). Basal processes of the anterior cell organize the anlage of the brain neuropile, while basal processes of the posterior cell do not go far from the cell body (Fig. 2L). Nuclear staining confirms that the positive signal occurs in the body of the anterior solitary cell but not in the neighboring cells (Fig. 2J).

By the end of the sixth day, the body of the anterior cell is no longer visible in the episphere (Fig. 2H). Despite the disappearance of the positive reaction in the early anterior cell, the tubulin-positive fibers organize the compact brain neuropile, ventral bundles, and first and second commissures (Fig. 2H). Two cells are located symmetrically at the base of the first and second commissures, and their processes join the ipsilateral ventral nerve bundle (Figs. 2H, 2K). The posterior cell is not visible from the ventral side because it is displaced more dorsally following the curved posterior region of the trochophore (Fig. 2H). However, the tubulin-positive cell body and its two short lateral processes can be seen on the dorsal side. Nuclear staining confirms that the positive signal occurs in the body of the anterior solitary cell but not in the neighboring cells (Fig. 2M). No 5-HT-IR and FMRFa-IR elements were detected (Fig. 2I).

Neurogenesis during the middle trochophore stage

The first specific 5-HT-IR and FMRFa-IR cells are detected at the middle trochophore stage in D. gyrociliatus. While only specific 5-HT-IR cells are detected in D. vorticoides. Note that the 5-HT-IR and FMRFa-IR processes from those cells constitute only a minor portion of the neural structures visualized with a pan-neural marker acetylated α-tubulin.

In D. gyrociliatus, tubulin-positive processes form a dense brain neuropile, paired ventral and lateral bundles, and unpaired medial bundle; two commissures connect ventral bundles; solitary thick processes run from the brain to the anterior region of the body (Figs. 3A and 3B). The early anterior cell remains in its position ventrally above the mouth. A single cell starts to express tubulin-positive immunoreactivity at the dorsal posterior region of the body; thus, it is not visible from the ventral side (Figs. 3C and 3D). However, a single tubulin-positive body of posterior cell can be distinguished among the tubulin-negative neighboring cells on the dorsal side (Fig. 3F).

Figure 3 5-HT-like immunoreactivity in the middle trochophore stage of Dimoprhilus gyrociliatus and Dinophilus vorticoides.

Yellow–acetylated tubulin immunoreactivity; magenta 5-HT-like immunoreactivity; red–DAPI. Ventral view. The apical pole is up. A–F–Dimorphilus gyrociliatus; G–N–Dinophilus vorticoides. (A) Tubulin-positive processes mark a neuropile (np), paired ventral (vb) and lateral bundles (ln), unpaired medial bundle (mn), and commissures (c); (B–D) Bodies of 5-HT-immunopositive cells (double arrows) are located at the base of first and second commissures; note 5-HT immunoreactive puncta in the commissure (c) and neural bundles. (E, E′) High magnification of 5-HT-like immunoreactive cell bodies at the base of commissure; (F) No 5-HT-like immunoreactivity occurs in the body of the posterior cell. (G) Tubulin-positive processes compose the compact neuropile (np), paired ventral (vb) bundles and commissures (c); (H, I) First 5-HT-like immunoreactivity appears in the cells (double arrows) at the base of first and second commissures (c); (K) No 5-HT-like immunoreactivity is detected in the neuropile (np); (L) Close up on three commissures (c) and 5-HT-like immunoreactive cells (double arrows); (M) High magnification of 5-HT-like immunoreactive body of a cell at the base of commissure; (N) No 5-HT-like immunoreaction is registered in the body of the early posterior cell. Abbreviations: act–acrotroch, n–nucleus, pt–prototroch. Scale bars: A–C, G–I–25 μm; D–7 μm, E–3.5 μm; E′, F–2 μm; K, L–5 μm; M–3.5 μm; N–2 μm.

Neither anterior nor posterior cells demonstrate a positive reaction to serotonin.

The first positive 5-HT immunoreaction occur in three cells located symmetrically at the left and right sides at the base of the first commissure (Figs. 3B–3D). Their long basal processes join the first commissure and the ventral bundles going posteriorly and anteriorly to the brain neuropile (Figs. 3B–3D). Nuclear staining confirms that each 5-HT-IR element represents a solitary cell–thus, two groups of three cells are located at the base of the commissure (Figs. 3D, 3E, 3E′).

D. vorticoides demonstrate a similar pattern of tubulin-positive structures as D. gyrociliatus with the exception of unpaired medial and paired lateral bundles while possessing more dense brain neuropile and one additional commissure (Figs. 3G and 3L). Note the absence of tubulin-positive bodies in the anterior region, while thick processes extending from the neuropile are clearly visible (Fig. 3K). The posterior cell is present on the dorsal side and demonstrates no positive reaction with 5-HT (Fig. 3N).

The first 5-HT-IR cells are found as groups of two cells located at the base of first commissure and solitary cells located at the base of second commissure (Figs. 3H, 3I and 3L). Nuclear staining confirms that the 5-HT-IR element at the base of the commissure represents a solitary cell (Fig. 3M).

FMRFamide-positive elements occurred at the middle trochophore of D. gyrociliatus but not in D. vorticoides. Staining was confined to the body and processes of the early anterior cell, visible at the previous stage after anti-tubulin staining (Fig. 4A). Specific staining allows us to visualize the processes of the early anterior cell among the tubulin-positive fibers. Numerous long processes extend basally from the cell body and run along with the paired ventral and lateral and unpaired medial bundles (Fig. 4B). The lateral basal processes of the early anterior cell first enter the brain neuropile, then go along the prototroch to the middle line of the embryo, then turn in the posterior direction and follow the lateral bundles (Fig. 4C). The short apical process runs toward the anterior surface (Figs. 4D′, 4E) and bears two long cilia (Figs. 4E and 4F). Nuclear staining confirms that only early anterior cells express positive FMRFa reaction in the anterior region (Figs. 4D, 4E and 5).

Figure 4 First FMRFa-immunoreactivity appears at the middle trochophore stage of Dimoprhilus gyrociliatus.

Yellow–acetylated tubulin immunoreactivity; cyan–FMRFa-like immunoreactivity; red–DAPI. A, B–Ventral view, C–view from the apical pole. (A) Tubulin-positive fibers form the main structures of the nervous system: neuropile (np), paired ventral (vb) and lateral bundles (ln), and unpaired medial bundle (mn), and commissures (c); (B, C) first FMRFa-like immunopositivity is located to the early anterior cell (asterisk) sending its processes to the neuropile, ventral, lateral and medial bundles; (C) Solitary FMRFa-immunoreactive anterior cells and its processes within the lateral bundles; (D) Nuclear labeling reveals a nucleus (arrow) surrounded by FMRFa-like immunopositivity, confirming that this immunoreactivity is located to the cell body; (D′) A short apical process (double arrow) of the first FMRFa-positive cell; (E, F) The Short apical process (double arrow) of the early anterior FMRFa-immunoreactive cell bears cilia (triple arrow). Scale bars: A–C–30 μm; D, D′–8 μm; E, F–5 μm.

Figure 5 The pathways of the first anterior FMRFa-positive cell processes in the middle trochophore stage of Dimorphilus gyrociliatus.

Yellow–acetylated tubulin immunoreactivity; cyan–FMRFa-like immunoreactivity; red–DAPI. Anterior ventral view, A–D–serial optical sections. (A) Surface section showing that the FMRFa-immunoreactive cell body (asterisk) is above the prototroch; (B) Section at the level of the neuropil shows that the FMRFa-immunoreactive cell (asterisk) is located anterior to the neuropile (np); (C, D) The lateral processes of the early anterior cell, innervate the prototroch and run in the posterior direction within the lateral bundles. Scale bar–25 μm.

Neurogenesis during the late trochophore stage

The specific 5-HT-IR and FMRFa-IR cells and their processes are detected at the late trochophore stage in both D. gyrociliatus and D. vorticoides. Still, 5-HT-IR and FMRFa-IR processes extending from those cells constitute only a minor proportion of the structures visualized with a pan-neural marker acetylated α-tubulin. Contrary to D. gyrociliatus FMRFa-IR cells appear in D. vorticoides at late trochophore stage only.

In D. gyrociliatus, tubulin-positive processes become more prominent. They constitute a dense brain neuropile, paired ventral and lateral bundles, and unpaired medial bundle; two commissures connect ventral bundles; solitary thick processes run from the brain to the anterior region of the body (Fig. 6A). The early anterior cell remains its position above the mouth. This cell bears a thin process running to the surface (Fig. 6D). Nuclear staining confirms that the positive immunoreaction occurs in the body of the single solitary anterior cell but not in the neighboring cells (Fig. 6D). A single posterior cell is present in the dorsal posterior region of the body (Fig. 6F). Neither anterior nor posterior cells demonstrate a positive reaction to serotonin.

Figure 6 The additional 5-HT-immunoreactive cells appear during the late trochophore stage in Dimoprhilus gyrociliatus and Dinophilus vorticoides.

Yellow–acetylated tubulin immunoreactivity; magenta–5-HT-like immunoreactivity, red – DAPI. A, B, G, H–Ventral view, the apical pole is up. C, I–Anterior top view. A–F–Dimorphilus gyrociliatus; G–M–Dinophilus vorticoides. (A) tubulin-positive processes become more prominent and constitute a compact brain neuropile (np), paired ventral (vb) and lateral bundles (ln), and unpaired medial bundle (mn); two commissures (c) connect ventral bundles. The early anterior cell remains its position (asterisk); (B) 5-HT immunoreactive cells (double arrows) located as two groups at the base of commissures, their processes follow the ventrolateral and medial bundles and commissures; (C) additional 5-HT immunoreactive cells (double arrows) in the anterior region above the head neuropile, and their processes ventrally to the longitudinal cords; (D) the early anterior cell does not express 5-HT like immunoreactivity; (E) close up on the ventral neural cord region; 5-HT immunoreactive processes are visible in the ventrolateral and medial bundles; (F) the early posterior cell are negative to 5-HT immunoreactivity. (G) tubulin-positive processes constitute the brain neuropile (np), paired ventral (vb) and lateral bundles (ln), and unpaired medial bundle (mn); three commissures (c) connect ventral bundles, (H) 5-HT immunopositive cells above the head neuropile (double arrows) and their processes in ventrolateral bundles. 5-HT immunoreactive cells at the level of commissures become more prominent. (I) apical view of four 5HT immunoreactive cells located above the neuropile. Note 5HT immunoreactive processes in the first commissure, (J) a close up of the head neuropile region; (K, L) High magnification of 5-HT-like immunoreactive cell bodies above the head neuropile (K) and at the base of the commissure (L); (M) No 5-HT-like immunoreactivity occurs in the body of the posterior cell. Scale bars: A–C, G–I–25 μm; D, E–10 μm; F–6 μm; K–2 μm; J–15 μm; L–8 μm.

The first positive 5-HT immunoreaction occurred in four cells located symmetrically at the left and right sides above the brain neuropile (Fig. 6C). Their long basal processes join the brain neuropile and follow the ventral bundles (Figs. 6B–6D). Two groups of 5-HT-IR cells at the base of first commissure become more prominent. Their processes are visible in the paired ventral and unpaired medial bundles (Fig. 6E).

D. vorticoides demonstrates a similar pattern of tubulin-positive structures as D. gyrociliatus (Fig. 6G). Note the absence of tubulin-positive bodies in the anterior region, while thick processes extending from the neuropile are clearly visible (Fig. 6J). The posterior cell is present on the dorsal side and demonstrates no positive reaction with 5-HT (Fig. 6M).

First 5-HT-IR cells found above the brain neuropile as two- four solitary cells (Figs. 6H and 6I). Nuclear staining confirms that the 5-HT-IR elements under the brain neuropile represent separate cells (Fig. 6K). 5-HT-IR elements at the level of commissures become more prominent as well as paired ventral and unpaired medial bundle (Figs. 6H, 6I and 6K). Nuclear staining confirms that each 5-HT-IR element represents a solitary cell (Figs. 6K and 6L).

Solitary anterior FMRFamide-positive cell and their processes in D. gyrociliatus become more prominent by late trochophore stage (Figs. 7A–7C, 7F).The early anterior cell demonstrates FMRFa-IR and its multiple processes can be visible in main nerve structures: the brain neuropile, paired ventral and lateral and unpaired medial bundles (Figs. 7B, 7C and 7F). The early anterior cell has short surface directed process (Figs. 7D and 7E). Nuclear staining confirms that only early anterior cells express positive FMRFa reaction in the anterior region (Figs. 7D and 7E).

Figure 7 FMRFa-immunoreactivity at the late trochophore stage in Dimoprhilus gyrociliatus and Dinophilus vorticoides.

Yellow–acetylated tubulin immunoreactivity; cyan–FMRFa-like immunoreactivity, red–nuclear staining with DAPI. A, B, G,–ventral view. H–dorsal view. C, I–lateral view. The apical pole is up. A–F–Dimorphilus gyrociliatus; G–M–Dinophilus vorticoides. (A) Anti-tubulin labeling reveals dense brain neuropile (np), paired ventral (vb) and lateral bundles (ln), unpaired medial bundle (mn), and three commissures (c). The early anterior cell (asterisk) remains its position above the mouth; (B, C) early anterior cell express FMRFa-immunoreactivity. Multiple FMRFa-immunoreactive processes visible in the brain neuropile (np), paired ventral (vb) and lateral (ln) and unpaired medial bundles (mn); (D) a close up of the early anterior cell with a surface directed process (arrow). Cilia (double arrows) located at the top of the short process; (E) Only early anterior cell express FMRFa immunoreactivity in the anterior region; (F) a close up of the FMRFa immunoreactive processes in ventrolateral, medial and lateral bundles. (G) tubulin-positive processes form a dense brain neuropile (np), paired ventral (vb) and lateral bundles (ln) and two commissures (c) connect ventral bundles; (H) First FMRFa-like immunoreactive cells above the brain neuropile; (I) an additional third FMRFa-like immunoreactive cell become visible a bit later. Solitary FMRFa-like immunoreactive cell located in the gut region; (J–L) FMRFa-like immunoreactivity is in the bodies of three head cells (J, K) and solitary gut cell (L); (M) No FMRFa like immunoreactivity occurs in the body of the posterior cell. Scale bars: A–C, G–I–25 μm; D–E, J, L–5 μm; F–8 μm; K–5 μm; M–2.5 μm.

First FMRFa-IR elements appear at the late trochophore stage in D. vorticoides above the brain neuropile (Figs. 7H and 7I). Nuclear staining confirms two solitary cells at the beginning of the late trochophore stage (Figs. 7H and 7J); an additional third cell is visible a bit later (Figs. 7I and 7K). One more solitary FMRFa-IR element appears at the level of intestine (Fig. 7I). Nuclear staining confirms that this element represents a cell (Fig. 7L). Note that intestine cell and the posterior cell are different, because the posterior cell does not express FMRFa at this stage (Fig. 7M).

The early anterior cell in D. gyrociliatus is present throughout the whole lifespan

The anterior FMRFa-IR cell in D. gyrociliatus is detected through the early, middle, and late trochophore stages, juvenile, adult, and even senior stages (Figs. 8A–8G). It is an unpaired cell located medially in front of the brain (Fig. 8D). This cell retains its position and morphology at all the aforementioned stages, including adult and senior (Figs. 8F–8F″, 8G–8G″). The cell body is 1.5 times bigger (20 ± 3 µm in diameter, n = 120) and can be easily distinguished among the other FMRFa-IR neighboring cells at senior stage (Figs. 8G, 8G′, 8G″). Prominent short thick apical processes protrude from the body surface and bear cilia at the anterior tip (Figs. 8A′–8C′, 8A″–8C″). The thin basal FMRFa-IR processes of this early anterior cell are visible among the dense brain neuropile (Figs. 8C, 8C″, 8D). Multiple basal processes can be followed in the prototroch nerve, ventral and lateral bundles (Fig. 8D).

Figure 8 The ontogeny of the early anterior cell in Dimorphilus gyrociliatus from the early trochophore stage to the elderly animal.

Red–acetylated tubulin immunoreactivity; cyan–FMRFa-like immunoreactivity. The early anterior cell is shown in frontal (A–G), apical (A′–G′), and lateral (A″–G″) projections from the early trochophore to elderly animals. (A–A″) Early trochophore stage. The solitary cell is located above the prototroch and bears cilia (double arrows). No positive FMRFa-immunoreaction was detected in the cell body. (B–B″) Middle trochophore stage. The early anterior cell starts to express FMRFa-immunoreactivity. (C–C″, F–F″) Late trochophore, juvenile, adult, and elderly animals stages respectively. Solitary early anterior cell located separately from the CNS neuropile (D, G–G″). Early cell retains its morphology and position at the juvenile stage. Scale bars: A–C, F–G–5 μm; D–40 μm.

It is worth repeating that in D. vorticoides, the anterior cell degenerates by the end of the early trochophore stage and never expresses FMRFa-IR or 5-HT-IR.

Detailed structure of the Dinophiliformia apical region

We incubated D. vorticoides in 5-HT and 5-HTP to test whether the apical cells are able to uptake serotonin or synthesize it from the immediate biochemical precursor. This feature is typical for cells belonging to the apical sensory organ (ASO) of most studied Lophotrochozoa.

In normal conditions, several flask-shaped cells are visible at the middle trochophore stage with anti-tubulin antibodies at the anterior region of D. vorticoides (Fig. 9B). Upon short-term incubation (3 h) in 5-HT (10−5 M) and 5-HTP (10−5 M), two of these tubulin-positive cells start to express positive 5-HT-IR (Figs. 9B′, 9B″). These paired cells are detected in the top anterior region, and the long basal process joins the brain neuropile (Figs. 9B′, 9B″). This experiment indicates the ability of certain apical cells to uptake exogenous serotonin as well as to convert 5-HTP to 5-HT. Note that cells with such a feature were detected at neither the early trochophore stage (Figs. 9A, 9A′, 9A″) nor later in the course of ontogenesis: late trochophore (Figs. 9C, 9C′, 9C″), in pre-hatching (Figs. 9D–9D″), juvenile (Figs. 9E–9E″), and adults (Figs. 9F–9F″), while distinct 5-HT-IR occurred within cell bodies adjacent to the brain neuropile (Figs. 9C–9C′) as well as within the brain neuropile and in the main neuronal bundles (Figs. 9C–9F, 9C′–9F′).

Figure 9 5-HT immunoreactive cells in the apical region after 5-HT and 5-HTP treatment from the early trochophore to the adult stage in Dinophilus vorticoides.

Magenta–acetylated tubulin immunoreactivity, green–5-HT-like immunoreactivity. (A–A″) During the early trochophore stage, no 5HT-immunoreactivity is visible in control (A), 5-HT-treated (A′), and 5-HTP-treated (A″) animals. (B–B″) During the middle trochophore stage, two apical cells are visible above the neuropile after 5-HT (B′) and 5-HTP (B″) treatment, while no 5HT-like immunoreactive cells are visible in the control group (B). (C–C″) During the late trochophore stage, apical immunopositive cells are detected neither in control (C) nor after 5-HT (C') and 5-HTP (C″) treatment. (D–F) No apical cells are visible in control and 5-HT or 5-HTP treated worms at the pre-hatch (D–D″), juvenile (E–E″), and adult stages (F–F″). Scale bars–40 μm.

Numerous epithelial cells with short sensory processes become 5-HT-IR in free-living pre-hatching, juvenile and adult D. vorticoides after short-term incubation in 5-HTP but not 5-HT (Figs. 9D″–9F″). Bunches of cilia are visible at the tip of anterior process in each cell (Fig. 9F″). This experiment demonstrates the presence of decarboxylase aromatic amino acid enzyme but not a serotonin transporter in some sensory epithelial cells.

Discussion

We provide a detailed analysis of the early neuronal development of D. gyrociliatus and D. vorticoides. We define three stages of embryonic development in both species and describe their ciliary and nervous structures landmarks (Figs. 10–14).

Figure 10 Schematic diagram of the development of tubulin-immunoreactive nerve system in Dimorphilus gyrociliatus and Dinophilus vorticoides.

The first nerve cells appear at the early trochophore stage in D. gyrociliatus and D. vorticoides. In D. gyrociliatus, it is the peripheral early anterior cell bearing surface cilia and located above the prototroch on the ventral side. This cell has fibers emanating from its basal part. In D. vorticoides, there are two early peripheral cells: anterior and posterior, both of which are non-sensory. These cells have multiple fibers. In D. gyrociliatus, the early anterior cell is present during the middle and late trochophore stages. By contrast, the anterior cell is detectable only during the early trochophore stage in D. vorticoides. In both D. gyrociliatus and D. vorticoides, the nervous system develops from both anterior and posterior poles of the body. The apical end is up. Relative dimensions are not to scale.

Figure 11 Schematic diagram of the development of 5-HT-immunoreactive nerve system in Dimorphilus gyrociliatus and Dinophilus vorticoides.

The first 5HT-like immunoreactive cells appear at the middle trochophore stage at the level of the first commissure in both species. The processes of these cells join the ventral bundles and the commissure. Additional 5-HT-like immunoreactive cells appear in the head region at the late trochophore stage. The apical end is up. Relative dimensions are not to scale. Blue–tubulin; pink–5-HT.

Figure 12 Schematic diagram of the development of FMRFa-immunoreactive nervous system in Dimorphilus gyrociliatus and Dinophilus vorticoides.

The first solitary FMRFa-like immunoreactive cell appears anteriorly at the middle trochophore stage in D. gyrociliatus. In D. vorticoides, two first FMRFa-like immunoreactive cells differentiate above the brain neuropile at the late trochophore stage. The apical end is up. Relative dimensions are not to scale. Blue–tubulin; salmon–FMRFa.

Figure 13 Schematic diagram of the neurogenesis in Dimorphilus gyrociliatus and Dinophilus vorticoides.

The early anterior cell in D. gyrociliatus and the anterior and posterior cells in D. vorticoides appear during the early trochophore stage. Initially, they do not exhibit5-HT-like or FMRFa-like immunoreactivity. Later, the early anterior cell in D. gyrociliatus becomes FMRFa immunopositive. 5-HT-like immunoreactive and FMRFa-like immunoreactive fibers represent only a tiny portion of the entire nervous system. The patterns of 5-HT-like immunoreactive look comparable, whereas the patterns of FMRFa-like immunoreactive are strikingly different in the two species. The apical end is up. Relative dimensions are not to scale. Blue–tubulin; pink–5-HT; salmon–FMRFa.

Figure 14 Comparative diagram of the early neurogenesis across Annelida tree.

The diagram is based on the recent data on Sedentaria (Kumar et al., 2020); Errantia (Starunov, Voronezhskaya & Nezlin, 2017); Dinophiliformia (this study); Sipuncula (Carrillo-Baltodano et al., 2019), and Oweniidae (Carrillo-Baltodano et al., 2021). The phylogeny data is retrieved from (Martín-Durán et al., 2021). Sedentaria, Errantia, and Dinophiliformia demonstrate the development of the nervous system starting from both the anterior and posterior poles (blue arrows). In Sipuncula and Oweniidae, neurogenesis proceeds from the anterior pole. The apical end is up. Relative dimensions are not to scale.

Since the duration of the embryonic period is different in D. gyrociliatus and D. vorticoides, we needed to validate the method, allowing to match their developmental stages. We found out that the pattern of differentiation of external ciliary structures is the same in both species: the prototroch appears first, followed by the ventral ciliary field and then by ciliary bands. Thus, these ciliary structures (along with an absolute timescale) can be used as temporal landmarks to compare developmental stages in two Dinophiliformia species with different durations of time they spend inside the egg. An external ciliation pattern was successfully used to define a particular developmental stage in other annelids (Bergter, Brubacher & Paululat, 2008).

Although Dinophiliformia develop directly, their early developmental stages are comparable with free-swimming larvae: early, middle and late trochophores. The trochophore is characterized by the presence of the prototroch, a preoral ciliary band (Rouse, 1999, 2000). Dinophiliformia retain this structure throughout their life as a paedomorphic trait (Kerbl et al., 2017). A recent analysis of gene expression confirmed the homology of the prototroch in indirect-developing annelids (Kempf, Page & Pires, 1997; Kerbl et al., 2016b) and the prototroch of adult D. gyrociliatus (Kerbl et al., 2017).

Comparative analysis of Dinophiliformia early neurogenesis

The early neurogenesis in many different lophotrochozoans has been thoroughly studied during the last few decades in order to understand the differentiation, evolution, and physiology of the nervous system in these animals. A considerable amount of this research was performed using a simple and reliable method of the visualization of nervous structures: immunostaining with antibodies to ubiquitously present neuronal markers: acetylated tubulin (pan-neural marker), serotonin (5-НТ), and a neuropeptide FMRFamide (Croll & Voronezhskaya, 1995, 1996; Conzelmann & Jékely, 2012). We used the same approach in our work in order to gain a picture comparable to other studies.

Dinophiliformia is a sister to Pleistoannelida, uniting Sedentaria and Errantia (Martín-Durán et al., 2021). We further discuss the early events in the neurogenesis of Dinophiliformia and compare them to other lophotrochozoan animals within and beyond this clade.

The neurogenesis in both Dinophiliformia begins with the differentiation of two peripheral cells, one anterior and one posterior, emanating from their processes toward each other. This type of localization of the first cells expressing neuronal characters was registered in other annelids from Sedentaria (Kumar et al., 2020) and Errantia (Voronezhskaya, Tsitrin & Nezlin, 2003; Starunov, Voronezhskaya & Nezlin, 2017). Some other representatives of Sedentaria, however, lack a posterior cell (Brinkmann & Wanninger, 2009; Meyer & Seaver, 2009; Meyer et al., 2015; Carrillo-Baltodano & Meyer, 2017), suggesting that the number and localization as well as transmitter content of the first cells in annelids can vary.

The processes of the early anterior and posterior cells in both Dinophiliformia run in the area of the prospective central nervous structures and create its general layout: brain neuropile, paired ventral and lateral and unpaired medial longitudinal nerve bundles and several ventral commissures. Nascent neural cells differentiate along the way of these processes. Such a scenario of neural development, when neurogenesis starts from two or more peripheral neurons located on the opposite poles of the larva, was considered ancestral for Pleistoannelida (Kumar et al., 2020). Our data on Dinophiliformia support this point of view (Fig. 14) and push back the presence of this type of neurogenesis to the common ancestor of Dinophiliformia and Pleistoannelida. Actually, this pattern of neurogenesis was also registered in other groups, for example, in mollusks (Redl et al., 2014; Pavlicek, Schwaha & Wanninger, 2018; Battonyai et al., 2018; Yurchenko et al., 2019), Nemertea (Hiebert & Maslakova, 2015), and Platyhelminthes (Rawlinson, 2010). This suggests that it might be much more ancient unless it evolved independently in these groups.

The fact that the processes of the early anterior and posterior cells in Dinophiliformia demarcate the layout of the central nervous system may indicate that these cells pioneer the development of the nervous system, although further molecular support is required. Whether these cells are pioneer neurons or not, the fact that they lack 5HT/FMRFa immunoreactivity set Dinophiliformia apart from many other Lophotrochozoa from distant groups, where the first neurons to differentiate are either 5HT or FMRFa immunoreactive. Representatives of different annelid groups (Oweniidae, Sedentaria, Errantia) show immunoreactivity to 5HT or FMRFa or both in their early anterior and posterior cells (Nedved, 2010; McDougall et al., 2006; Starunov, Voronezhskaya & Nezlin, 2017; Voronezhskaya, Tsitrin & Nezlin, 2003). Non-annelid Lophotrochozoa (mollusks) demonstrate FMRFa immunoreactivity in the presumable pioneer neurons (Voronezhskaya et al., 1999; Dickinson, Croll & Voronezhskaya, 2000). Aside from this, 5HT or FMRFa immunoreactivity was detected in the first neural cells, associated with different structures and functions, in nemerteans (Chernyshev & Magarlamov, 2010; Von Döhren, 2016), flatworms (Rawlinson, 2010), mollusks (Dyachuk & Odintsova, 2009; Pavlicek, Schwaha & Wanninger, 2018; Battonyai et al., 2018; Yurchenko et al., 2019), phoronids (Hay-Schmidt, 1990), and brachiopods (Altenburger & Wanninger, 2010). The ubiquitous presence of 5HT or FMRFa immunoreactive early neurons in Lophotrochozoa suggests that it is likely to be an ancestral state. However, there are a few exceptions: a sedentarian annelid M. fuliginosus (Kumar et al., 2020), and some mollusks (Redl et al., 2014) possess the early neuron with other (still unidentified) transmitter phenotypes. Certainly, the first neurons do not have 5HT and FMRFa immunoreactivity in these animals. Thus, the absence of 5HT and FMRFa immunoreactivity in the first neurons is not unique for the early stages of Dinophiliformia but is apparently rare for annelids as well as other lophotrochozoans, and may suggest a derived state of early neurogenesis. Further research is needed to identify signaling molecules expressing in the early cells of Dinophiliformia.

The anterior cell in D. gyrociliatus is a particular case

The first neurons of Lophotrochozoa, which could be either pioneer neurons or sensory cells of the apical organ or other cells with an unknown function, are known to be transient as they disappear by the end of larval development (Voronezhskaya & Ivashkin, 2010; Nezlin & Voronezhskaya, 2017). Although it was shown in D. vorticoides, the anterior and posterior early cells in D. gyrociliatus are retained throughout the whole life of the species. At the early stage, these cells are detected with anti-tubulin antibodies; later on, they begin to express FMRFa immunoreactivity. The early anterior cell of D. gyrociliatus has a short apical neurite and several cilia and thus can be considered a sensory cell. In addition, this cell sends long basal processes to the brain neuropile, the prototroch circular nerve, and the ventral, medial, and lateral cords. It does not seem to be homologous to the anterior cell of D. vorticoides, which does not have FMRFa immunoreactivity and is not sensory since it has no surface extensions. The fact that the early anterior and posterior cells of D. gyrociliatus are present throughout the entire lifespan may be interpreted as a paedomorphic trait in D. gyrociliatus, related to its progenetic origin (Kerbl et al., 2016a).

The FMRFa-immunoreactive cells with similar location and/or time of appearance have been registered in close relatives of Dinophiliformia, C. teleta (Sedentaria) (Meyer & Seaver, 2009; Meyer et al., 2015; Carrillo-Baltodano & Meyer, 2017), P. maculata (Errantia) (Voronezhskaya, Tsitrin & Nezlin, 2003), and P. agassizii (Sipunculida) (Kristof, Wollesen & Wanninger, 2008). It has not been registered in other annelids and other Lophotrochozoa groups. Although these FMRFa-immunoreactive cells are not the first to differentiate in the aforementioned species, it is plausible that they are homologous to FMRFa-immunoreactive cell in D. gyrociliatus. Therefore, this cell might be a synapomorphy of a clade Pleistoannelida + Dinophiliformia + Sipuncula (Fig. 14).

Where is the apical organ?

The apical organ is a conservative neural structure developing early during the neurogenesis of lophotrochozoan larvae. It is a group of sensory FMRFamide- or 5-HT-immunoreactive cells, located adjacent to cerebral commissure and transient in appearance. The apical organ is thought to have been present in the common ancestor of Lophotrochozoa (Nielsen, 2005; Wanninger, 2008; Marlow et al., 2014) or even the common ancestor of Cnidaria and Bilateria (Marlow et al., 2014), and thus represents a plesiomorphic state in these animals. Our study on Dinophiliformia revealed no cells which could be assigned to the apical organ by their expression of 5-HT or FMRFamide immunoreactivity. However, anti-tubulin antibodies revealed sensory flask-shaped cells in an anterior region of embryos of D. gyrociliatus and D. vorticoides. The pharmacological approach showed that two of these cells could uptake 5-HT and synthesize it from the precursor during the limited period of neurogenesis. This result indicates that these cells have the transient expression of enzymes necessary for serotonin synthesis and transporters using for 5-HT uptake, and could thus be a remnant of an ASO cells. Therefore, the apical organ of Dinophiliformia was either significantly reduced or switched to the use of other signaling molecules, for example, other than FMRFamide neuropeptides, a wide variety of which are highly expressed in the brain of adult Dinophiliformia (Kerbl et al., 2017) and present in the apical sensory cells of other annelids (Kerbl et al., 2017). Both scenarios could result from Dinophiliformia evolution, which included simplification, miniaturization, paedomorphosis, and switch to direct development (Westheide, 1987; Hanken & Wake, 1993; Müller & Westheide, 2002; Kerbl et al., 2016a, 2016b).

Free-swimming annelid larvae usually have a prominent apical organ (Voronezhskaya, Tsitrin & Nezlin, 2003; McDougall et al., 2006; Starunov, Voronezhskaya & Nezlin, 2017; Kumar et al., 2020; Carrillo-Baltodano et al., 2021). However, it was suggested to be reduced in a sedentarian C. teleta (Meyer & Seaver, 2009; Meyer et al., 2015; Carrillo-Baltodano & Meyer, 2017). Given that the trochophores of another sedentarian (Kumar et al., 2020) and Errantia (Voronezhskaya, Tsitrin & Nezlin, 2003; Starunov, Voronezhskaya & Nezlin, 2017) do have the apical organ, we speculate that the reduction of the apical organ in C. teleta and Dinophiliformia represents independent evolutionary events.

Conclusions

We revealed that the development of the nervous system in Dinophiliformia starts from two neural cells located at the opposite poles of the body and probably serve as pioneer neurons. This pattern of early neurogenesis is comparable to that of most representatives of Pleistoannelida and perhaps represents a synapomorphy of a clade Dinophiliformia + Pleistoannelida. However, Dinophiliformia has a few distinctions from most Pleistoannelida and other Lophotrochozoa in that their early anterior and posterior cells lack 5-HT or FMRFa immunoreactivity and the apical organ is reduced or simplified. Although Dinophiliformia share these characteristics with a few representatives of Pleistoannelida, we hypothesize that they evolved independently during the paedomorphic evolution of this group. We suggest that FMRFa immunopositive anterior cell is a synapomorphy for a clade Pleistoannelida + Dinophiliformia + Sipuncula. The fact that D. gyrociliatus, unlike D. vorticoides, retains this neuron through the lifespan probably reflects an extreme degree of paedomorphosis of this species.

These data add to our knowledge of how variable the early development in Annelida and Spiralia could be and to what extent evolutionary changes might affect such a conservative process as early neurogenesis.

The research was done using the equipment of the Core Centrum of the Institute of Developmental Biology RAS. Light microscopy study was conducted using equipment of the Center of microscopy WSBS MSU. We thank the anonymous native speakers from the Flarus and MDPI agencies for the professional language proofreading. The work was conducted under IDB RAS GBRP # 0088-2024-0001.

Additional Information and Declarations

Competing Interests

Author Contributions

Data Availability

The authors declare that they have no competing interests.

Elizaveta Fofanova conceived and designed the experiments, performed the experiments, analyzed the data, prepared figures and/or tables, authored or reviewed drafts of the paper, and approved the final draft.

Tatiana D. Mayorova conceived and designed the experiments, performed the experiments, analyzed the data, prepared figures and/or tables, authored or reviewed drafts of the paper, and approved the final draft.

Elena E. Voronezhskaya conceived and designed the experiments, analyzed the data, prepared figures and/or tables, authored or reviewed drafts of the paper, and approved the final draft.

The following information was supplied regarding data availability:

The raw data is available at figshare: Fofanova, Elizaveta; Mayorova, Tatiana D.; Voronezhskaya, Elena (2021): Raw data.rar. figshare. https://doi.org/10.6084/m9.figshare.16441293.v1.

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
