# Peer review of "Dinophiliformia early neurogenesis suggests the evolution of conservative neural structures across the Annelida phylogenetic tree"

_PeerJ, doi:10.7717/peerj.12386_

## Round 0.1 · original submission · Major Revisions

· Academic Editor

Major Revisions

Both expert referees appreciate the quality of this new data and insights that come from it, but both have a number of recommendations for improvements to the manuscript, which should all be accommodated as much as possible in order to greatly strengthen this paper.

Reviewer 1 ·

Basic reporting

1. Basic reporting: The manuscript is devoted to development of the nervous system of tiny dinophilid annelids. The organization and development of the nervous system are traditionally used for reconstruction of phylogeny. Therefore results of this report are important for future comparative analysis.

Experimental design

2. Experimental design: All experiments are correct and are consistent with methods of immunocytochemistry and confocal laser scanning microscopy.

Validity of the findings

3. Validity of the findings: All new results are correct and supported by original figures.

Additional comments

4. General comments:
Authors represent very interesting new results, which are well illustrated. These results are worth to be published. At the same time, the manuscript has to be improved and some corrections are necessary.
Major comments:
I kindly ask authors to discard the misunderstanding concerning one or two pioneer neurons in two different dinophilid species. I also ask authors to explain the term “scaffold”. And the third important note concerns the Figures. The combination of red and green colors in one figure is impoliteness to people who have problems with vision. I recommend authors to avoid such a combination of colors.
I recommend to invite native speaker to edit the English.

Minor comments:
Abstract: Using ICH and LSCM, we revealed that in both species, the first neurons appear at the most anterior and posterior regions of the embryo and revealed positive reaction with anti-alpha-acetylated tubulin antibodies only (no positive signal with anti-serotonin (a5-HT) or antiFMRFamide (aFMRFa) antibodies were detected).
--- I recommend authors to escape the use of unexplained abbreviations and repetitions.
--- Accordingly to text, Dimorphilus gyrociliatus lacks posterior perikarya (perikaryon). Thus, this phrase is incorrect and has to be rewritten.
Abstract: On the contrary, both anterior and posterior early cells in D. gyrociliatus maintain during the whole life, which probably reflects the progenetic trait of this species.
---- It sounds wrong. It means that life reflects progenetic trait. It is incorrect.
---- I did not understand this phrase, because Dimorphilus gyrociliatus lacks posterior perikarya (perikaryon). Thus, it is impossible to maintain both posterior and anterior cells if posterior cells are absent.
Abstract: The nervous system develops in the rostrocaudal direction in both species.
---- This state contradicts the early information in the abstract “Long neurites of early neurons grow towards each other…..”, I did not understand, how posterior neuron is able to give rise to neurites in rostrocaudal direction.
Abstract: Based on our data, we infer how the evolutionary history of dinophilids may have affected the early neurogenesis of these extraordinary animals.
----I recommend authors to briefly describe exactly conclusion instead of this phrase.
Introduction: In polychaetes and nemerteans, pioneer neurons demonstrate a positive reaction to serotonin antibodies (5-HT-IR) (Voronezhskaya and Elekes, 2003; Voronezhskaya and Khabarova, 2003; Voronezhskaya, Tsitrin and Nezlin, 2003; Chernyshev and Magarlamov, 2010; Fischer, Henrich, Arendt, 2010)
---- The citation of this paper (Heibert, Maslakova, 2015: EvoDevo) has to be added
Introduction: The apical organ consists of a group of flask-shaped cells whith short processes reaching surface of epithelium and bearing cilia on their apices.
--- Incorrect word
Introduction: Our results demonstrate that the early neurogenesis is similar in twodinophilid species but significantly different from that of other Lophotrochozoa.
--- I recommend delete this phrase from the Introduction.
Mwthods: ………..with diatom Phaeodactylum sp. and ………….
---- Use Italian
Methods: Both dinophilidspecies laid cocoons regularly.
----- Add space
Methods: ……..and six-well plates at 10°C in filtered seawater…………
--- I did not understand this
Methods: Immunocytochemistry
----Please indicate whether you blocked nonspecific staining. If yes, please describe how you processed
Methods: D. vorticoides at different developmental stages were incubated in 10-5 M 5-HT and 5-HTP (Sigma-Aldrich, USA) for 3 hours.
---Please explain all abbreviations before using.
Results: Line-249 Long processes of these cells construct the scaffold of the definitive nervous system. Line-254-255: As the number of this cell's processes increases, they spread farther and establish the scaffold of the definitive central nervous system.
---- Please delete repetitions.
Results: By the end of the prototroch stage, a pair of additional neurons differentiate at the…….
----Replace to differentiates
Results: Their processes extend along the scaffold formed by the first cell (Fig. 2 b, e). Thus, the entire nervous system's scaffold is built by the processes of the first single cell (Fig 2. b, e, d).
---- I do not see the “scaffold’ in Figures 2b, e, d. What do you mean as “scaffold” ? The same question concerns D. vorticoides
----Point has to be deleted
Discussion: Then establishes the larval nervous system, and after that the definitive nervous system differentiates
---Please check this phrase. It seems to be wrong
Discussion: Our results show that two first cells (the anterior and posterior ones) emerging at the early trochophore stage in two dinophilids (Fig. 9-14)
---- Dimorphilus gyrociliatus lacks posterior perikarya (perikaryon).
Discussion: The first neuron in D. gyrociliatus appears at the early trochophore stage at the anterior extreme of the embryo at the midline above the prototroch and is visualized with anti-tubulin antibodies only. The cell has a short apical neurite and several cilia…..
---This description has to be placed in Results. I recommend to indicate apical neurite and cilia by arrows in the Figure. I did not find any cilia in Figure 2a, b, which is devoted exactly to this neuron.
Discussion: The cell has a short apical neurite and several cilia (Fig. 3, Fig. 8, Fig. 10-13), and thus can be considered a sensory cell…..
--- as?
Conclusion: The early anterior neuron in D. vorticoides is transient, while it is detectable through the whole life in D. gyrociliatus, suggesting a paedomorphic trait of the later.
---This phrase means that only D. gyrociliatus, but not D. vorticoides has paedomorphic trait. I recommend rephrase.

Annotated reviews are not available for download in order to protect the identity of reviewers who chose to remain anonymous.

Reviewer 2 ·

Basic reporting

The manuscript cited above is a through description of the neurogenesis in two dinophilid species, one dimorphic with dwarf males and a monomorphic species: Dimorphilus gyrociliatus, and Dinophilus vorticoides. The study is well done and the authors provide a wealth of new data. On first sight, it appears astonishing to have another paper on these animals since they belong to the best studied within annelids fascinating researchers since their first discovery in the19th Century. There may be several reasons for a still ongoing scientific interest in these animals. One reason might be that one of the species is sexually extreme dimorphic possessing dwarf males, which exhibit an organization very different from that of the females. Other reasons might be that they formerly were regarded to exhibit ancient annelid structures (the Archiannelida concept) and D. gyrociliatus is very easy to culture, has short generation times making the species suitable for various kinds of experiments and last not least the adults are microscopic and can be studied as whole mounts under the microscope. Nevertheless, this study nicely complements our knowledge on early development in these creatures and allows some deductions to annelids and lophotrochozoans in general. The manuscript cites the relevant literature on the topic some minor problems are indicated below. Figures are necessary are well-balanced and well organized. However, I think the English definitely needs some revision by a native speaker.

Experimental design

The experimental design is clear; data are obtained using immune labeling of selected structures and confocal laser scanning microscopy, standard methods to address such scientific problems. However, I think the methods should be described more precisely. E.g., we don’t know the supplier or its place of the biochemicals, especially the antibodies, which might be crucial for repetition. Also giving two references for the protocol may not be the best solution rather than giving incubation times etc. This will not occupy too much space and the reference to previous work may still be justified. Giving the coordinates of the locality like it is done is useless because it is too imprecisely.

Validity of the findings

The title and certain phrases in the abstract and introduction may suggest that the authors doubt the placement of Dinophilidae and suggest some principle results for Lophotrochzoa in general but this leaves the reader somehow disappointed, because no such interpretations are given. In general certain points are just statements and should be explained by adding further information, references etc. This will be specified below.
I have some problems with identifying of the pioneer neurons by staining against acetylated tubulin only: As the authors know tubulin signal mainly comes from the neurites and some spherical staining is usually interpreted as soma without providing a counterstain that a nucleus is involved. As for instance shown in the paper by Kuma et al. (cited in the MS) the area of the nucleus is always recognizable by a spherical region without any immune-signal, which would be another proof for a nucleus. From the figures provided, I cannot follow the authors by their interpretation. They may be correct but evidence should be given by the data shown. Likewise, I cannot follow the author’s interpretation of the fate of the pioneer neurons. The authors provide a relative staging of the development in the two species in order to make the different speed of development comparable. These stages are given more neutral names but later on the author speak about a trochophore stage which is misleading in two points. First, the reader does not exactly know which stage is meant and, second and more importantly, this naming implies that this stage represent s a trochophore larva rather than an embryo, which it is really is since there is no larva (line 455 “is similar in two dinophilid trochophores” “other polychaete larvae”).
The findings should be discussed more properly and especially the paper of Kumar et al (2020) and Meyer and Seaver (2009) should be discussed and compared more carefully in order to determine common traits in annelids. Moreover, other studies used gene expression patterns to identify clearly pioneer neurons rather than by pure tubulin staining. Therefore, the authors should discuss their findings with care since some evidence is not provided. In addition, some findings are not properly documented by the authors; however, I am sure that the authors are able to provide missing evidence by their figures.

Additional comments

1. The title should express to which group of lophotrochzoans the animals of the present stud y belong.
2. In the abstract, the first two sentences say: “Despite the increasing data about neurogenesis in various Lophotrochozoa groups, the basal principles of the organization of their nervous system are still underestimated. Comparative data about the early developmental events from previously neglected taxa, non-model species, and rare groups are necessary to complete our knowledge.” For the abstract this is ok but in the text I am looking for justification for these statements I would like to know why we ned these data; do we have indications that the picture is more diverse than it would appear using only model organisms? Then it should be given.
3. In the abstract it is said that dinophilids are unique in combining “traits of different Lophotrochozoan taxa”. Without mentioning these traits and taxa this does not tell us much.
4. ICH and LSCM are not explained, neither in the abstract nor in the main text. I think LSCM means confocal laser scanning microscopy and to my knowledge CLSM is the abbreviation which is in common use.
5. Reading the introduction leaves somewhat helpless and puzzled. The first paragraphs appear to be rather long but mainly comprise references. The first sentence in lines 48/49 ist followed by more than 10 lines with references validating a mild context: “Over the past decades the research of lophotrochozoan neurogenesis and functions of early larval neurons has made considerable progress.” I wonder whether it might be better to cite a selection of these papers in order to make it readable. Alternatively more information may be provided telling us which refrence refers to which lophotrochzoan group. Moreover, I am not sure whether Struck 2006 and Richter et al. 2010 are cited in the correct context here. The next sentence almost tells us the same and again provides us with numerous references some of which appear for the second time.
6. Line 62: Polychaeta has been not to represent a taxon and as such the affiliation Polychaeta should be avoided. In case the authors do not agree with this grouping this should be justified and expressed.
7. In line 88 the authors introduce the apical organ without explaining its context: “The apical organ consists of a group of flask-shaped cells”. Information is lacking what an apical organ is and that it is developed in larvae of lophotrochozoans. Here also information is lacking that the species under study do not have such a larva.
8. Inline 111/12 the following statemante is given” for example, some structures might be completely lost or highly elaborated.” For me these examples are not given and I have to look for them myself in the literature. I think these should be added.
9. Line 115ff.: “For example, the nervous system of Phoronida and Brachiopoda demonstrate similarities with deuterostome neural systems (Nielsen, 2002; Nielsen, 2004; Nielsen, 2005; Temereva and Wanninger, 2012), doubting the molecular phylogeny view, which considers phoronids and brachiopods as Protostomia (Nielsen, 2002).” I think phylogenetic analyses of molecular data helped to resolve many open questions in metazoan phylogeny and this sebntences requires a statement what the authors think about this. On the one hand they apply the name Lophotrochozoa which originates from molecular systematics and if they question this they should express and explain this.
10. Line 127: “…Dinophilidae family (Archiannelida, Polychaeta) …” For many years now most annelid researches follow the hypothesis that a taxon Archiannlida is artificial and does not exist. If the authors want to re-introduce it here, this has to be justified and discussed. Currently there two hypothesis discussed, both of which place this group among typical polychaetes and without other former archiannelids.
11. Lines 132ff.: “Further investigations confirmed an unusual morphology of dinophilids: they combine characteristics of different lophotrochozoan. For example, they are similar to Polychaeta due to segmented epithelial structures, though have no chaeta [sic: chaetae]; parenchymatous organization and protonephridia make them similar to Platyhelminthes; they utilize gliding ciliary locomotion like mollusks …” This shall explain that dinophilids show traits of other lophotrochzoans. I see it as an assemblage of some outdated statements on invertebrate morphology and phylogeny. For instance, it is known for quite a long time that a number of annelids don’t possess coelomic cavities. Nevertheless their relationship has not been doubted. Moreover if one excepts that Annelids have a biphasic life cycle comprising a planktonic trochophore larva and a benthic adult it is clear that both types of body cavity organization belong to the ground pattern of Annelida: the larva is acoelomate and the adult is coelomate. Since body cavities and nephridia are functionally related to one another protonephridia are to be expected, if there is no coelom and blood vascular system. So this character may also occur in free-living Platyhelminthes, but if a relationship should be exist, the protonephridia of Dinophilidae should show the same apomorphies than those of Platyhelminthes, which is not the case. Ciliary gliding by means of ciliated epithelia is rather common in meiobenthic invertebrates and as such not all a character giving a hint to mollusk similarity. This assemblage looks like creating scientific questions were they don’t exist.
12. Line 154/5: “immunochemical staining with a pan-neural marker (anti-acetylated α-tubulin antibodies)” I wonder whether the authors are aware of the fact that all stabilized microtubules are stained with this antibody which may be more than nerve cells. This should be mentioned.
13. Line 164/5: “Water was changed every day to collect all developmental stages of embryos.” For me it is not clear why exchange of water makes sure to collect all embryos. To my knowledge the so called cocoons are laid on the substrate and thus one must either collect the cocoons or the adult animals and put them into fresh vials.
14. Line 261: “Their processes extend along the scaffold formed by the first cell (Fig. 2 b, e). Thus, the entire nervous system's scaffold is built by the processes of the first single cell (Fig 2. b, e, d). I cannot follow the description here why is it formed by the first cell when it is followed by two additional neurons? Please reword.
15. Line 278: “pan-neural marker“ this has been mentioned and explained in the introduction but should be mentioned in the M & M section as well to avoid confusion.
16. Line 282: “however, no neurons are detectable in the neuropil region …” Why do you use “however”? With some basic knowledge in annelid neuroanatomy you should know that neuropil and somata are always separated – if not the nervous system is medullary (see Richter et al. 2010!!).
17. Line 286: “In the caudal region, …” I think one should use morphological terms properly. To my knowledge, Dinophilidae do not possess a cauda rather than any tagmata. Strictly speaking, it is the locomotory organ of chordate and especially a primary aquatic vertebrate. So please use anterior or posterior, for instance. Likewise “head” may be better replaced by “prostomium”.
18. Line 298: “…and bearing a tuft of cilia … “ I could not see the cilia, please provide a figure.
19. Line 300/01: “The neuropile becomes larger; it is constituted by the neural processes only …” Again neuropil means that it only comprises neurites and synapses so the second part of the sentence is superfluous.
20. Line 308/9: “These neurons do not have cilia (Fig. 3 k).” Why should they have cilia?
21. Discussion, first paragraph: This si a repetition of the results and contains only extremely generalized statements which do not help to learn something new. This paragraph should be rewritten.
22. Line 439: Did Purschke (2005) show the nervous system of Dinophilidae in that paper?
23. Line 447ff.: “Therefore, the main differences between neurogeneses [sic: neurogenesis] in dinophilids and other lophotrochozoans lie at the very early events of neural development: appearance of pioneer neurons, which do not contain5HT and FMRFamide and in case of D. gyrociliatus are not transient (see below).” Some sentences above the authors say that this has also been observed in another annelid (Malacoceros). Is Malacoceros not an annelid and a lophotrochozoan? And Dinophilus? What are other lophotrochozoans? So, you have found a character common for Annelida which by the way are representatives of Lophotrochozoa. Where is the problem?
24. Line 475: “.he cell has a short apical neurite and several cilia (Fig. 3, Fig. 8, Fig. 10-13), …” Ok but I failed to find the cilia on the figures cited.
25. Reference 53 should not be in revision any more, please provide current data.
26. References in the text should be given without initials of authors (e.g. line 90/91)
27. Reference Hessling and westheid should read Hessling and Westheide (Line 126)
28. Line 132: I doubt whether Richter et al is cited correctly here;, please check.
29. Line 157 space missing: “twodinophilid”
30. Line 172: “Phaeodactylum” in italics
31. Line 175: space missing “dinophilidspecies“
32. Line 228: typo: „mouse“
33. Line 379: typo: “Paulat” should be “Paululat”
34. The list of reference should be reworked thoroughly, species names are frequently not in italics and there are several typos; e.g.: line 588 “gastropodIlyanassa obsolete (space missing, no italics).

---

## Round 0.2 · accepted · Accept

· Academic Editor

Accept

Thank you for carefully addressing all of the reviewers' comments.

At the proof stage please be sure to carefully check for typos, as reviewer 1 highlights that there are still some errors present.

Reviewer 1 ·

Basic reporting

The manuscript is considerably improved and is about ready for publication. Authors have processed all my comments and corrected the text and figures.

Experimental design

Experimental design is correct

Validity of the findings

Findings are interesting and valid.

Additional comments

Although the paper is greatly improved, I strongly recommend to make the figure legends more friendly for reader. In general, the figure with legend should be absolutely self-sufficient. Thus, I recommend authors to use full name of all species, which are mentioned in the figure legend. The full manes should be used at least in the title of the figure legend.
There are some typos in the text and figure legends. For example, Figure 14 "Phylogeny is base on....". I recommend authors to reread the text and correct the typos.

Reviewer 2 ·

Basic reporting

The MS under consideration is the reviesed version of a previous manuscript. Both reviewer raised a number of points which have been adressed by the authors.
My focus was on those points raised by me (reviewer #2).
Since the general evaluation of the mansucript was positive in spite of several points for improvement, I don't want to repeat the everything of this first evaluation.

Experimental design

The points raised for the experimental design have been sufficiently addressed.

Validity of the findings

The points raised in this part have been sufficiently addressed by the authors as well.

Additional comments

I don't see any points for further revsion.